# Perceptions, Knowledge, and Practices Concerning Indoor Environmental Pollution of Parents or Future Parents

**DOI:** 10.3390/ijerph17207669

**Published:** 2020-10-21

**Authors:** Laure Daniel, Marylou Michot, Maxime Esvan, Pauline Guérin, Guillaume Chauvet, Fabienne Pelé

**Affiliations:** 1Department of General Medicine, Rennes University, F-35000 Rennes, France; laurefdaniel@gmail.com (L.D.); mary2906@hotmail.fr (M.M.); pauline.guerin28@gmail.com (P.G.); 2CIC 1414 (Centre d’Investigation Clinique de Rennes), Rennes University, CHU Rennes, Inserm, F-35000 Rennes, France; maxime.esvan@chu-rennes.fr; 3ENSAI, CNRS, IRMAR-UMR 6625, Rennes University, F-35000 Rennes, France; guillaume.chauvet@ensai.fr

**Keywords:** indoor environment, perinatal health, knowledge, perceptions, practices

## Abstract

Indoor pollutants can have short- and long-term health effects, especially if exposure occurs during prenatal life or early childhood. This study describe the perceptions, knowledge, and practices of adults concerning indoor environmental pollution. Adults of 18 to 45 years of age were recruited in the department of Ille-et-Vilaine (Brittany-France) in 2019 through a stratified random draw in the waiting rooms of general practitioners (GPs) (*n* = 554) who completed a self-questionnaire. The 71% who had already heard of this type of pollution were older (*p* = 0.001), predominantly women (*p* = 0.007), not expecting a baby (*p* = 0.005), and had a higher knowledge score (*p* < 0.001). The average knowledge score was 6.6 ± 6.6 out of 11, which was higher for participants living in a couple and with a higher level of education (*p* < 0.001). Some practices were well implemented (>80% of participants) (aeration during renovation) whereas others were insufficiently practiced (<60% of participants) (paying attention to the composition of cosmetic products). Factors associated differed depending on the frequency of integration: living in a couple and having a child for well implemented practices and educational level, knowledge level, and perception for those under implemented. Knowledge must be improved to modify perceptions and certain practices, making sure not to increase social inequalities in health.

## 1. Introduction

A large body of evidence suggests that early prenatal or postnatal environmental exposure may influence the future health of children, adults, and even the geriatric population [1]. There is strong evidence concerning the impact of certain environmental pollutants such as outdoor air pollutants, heavy metals (lead, mercury, cadmium, arsenic), organochlorine compounds (polychlorinated biphenyls (PCBs), dichlorobiphenyl-trichloroethane/dichloroethylene (DDT/DDE), hexachlorobenzene (HCB)), or organophosphates insecticides on fetal growth and neurobehavioral development of infants as well as their respiratory and immune health, if exposure to these pollutants occurs during pregnancy or the first years of life [2]. Environmental exposure to pollutants during pregnancy and early childhood is also thought to play a role in the occurrence of cancers, cardiovascular disease, metabolic diseases, and reproductive disorders in adulthood [3,4,5,6]. Studies show that such disorders can occur as a result of exposure at much lower levels than those observed in adulthood [7,8] as developing organs and systems (from fetal life to childhood) are highly sensitive to their environment.

Indoor environment is a mixture of physical, chemical, and biological pollutants that originate from outdoor air, building and decorative materials, combustion appliances, and human activity [9]. Exposure to indoor environmental pollutants in the general population is ubiquitous, multiple, and chronic [10,11,12,13,14,15,16]. The routes of exposure and penetration into the body are ingestion, inhalation, dermal, mucosal, and transplacental. Pollution of the domestic environment is a public health issue because of the amount of time spent by the entire population in their homes (a mean of 16 h per day) and the wide variety of contaminants present [9]. However, for the French, pollution of the domestic environment is often wrongly perceived as being less risky than outdoor air pollution [17].

In France, health or environmental agencies and health professional representative agencies propose recommendations to improve the domestic environment by numerous means. These include prevention websites, brochures and information guides as well as media coverage [18,19,20,21,22,23,24]. A paragraph entitled “Advice for a healthy environment” was introduced to the new health booklet put into circulation in April 2018, encouraging parents to reduce the exposure of infants to sources of environmental pollution [25]. These recommendations concern many areas of daily life including food, hygiene products, cosmetics, air, textiles, household products, furniture, decoration, water, and toys and include advices such as ensuring sufficient ventilation, eliminating house dust, maintaining combustion appliances, limiting exposure to volatile organic compounds, monitoring materials containing asbestos, protecting children from lead paints, and controlling hot water contamination. 

Some studies have evaluated the perceptions, knowledge, and practices regarding the indoor pollution of parents or future parents. In the United States, a study has shown that the more parents are concerned about exposure to environmental chemical compounds, the more their children’s exposure to these compounds is reduced [26]. In France, most studies have focused on the perceptions [27,28,29,30] or knowledge [27] of pregnant women. A study carried out in 2015 in two French departments evaluated the practices of 128 women of childbearing age (18–45 years) (for a total of 60 non-pregnant women and 68 pregnant women) concerning their use of cosmetics (personal care products: hygiene and make-up) outside of and during pregnancy [31]. The “Pesti home” study conducted in France in 2014, provided an overview of the practices, uses, and determinants of uses of pesticides by French people (18–79 years) in their homes (homes, gardens, pets) [32]. However, no study has examined the perceptions, knowledge, and practices of parents or future parents in France of multiple type of exposure sources.

As a large proportion of the general population consults a general practitioner (GP) at least once a year [33] and because primary-care physicians intervene at a privileged moment to prevent environmental risks to child health, our objective was to describe the perceptions, knowledge, and practices of parents or future parents consulting with primary care physicians concerning pollutants in their domestic environment.

## 2. Materials and Methods 

### 2.1. Study Population

This cross-sectional descriptive study included 554 people aged 18 to 45 years. Participants were recruited in waiting rooms of GPs in the department of Ille-et-Vilaine (Britany, France) between 29 April 2019 and 6 December 2019 using a three-degree stratified random survey.

The three-degree stratified random study was designed as follows. For the first degree, a sample of GPs stratified according to the level of urbanization of the city in which they worked, was selected from the French shared directory of healthcare professionals (RPPS: Répertoire Partagé des Professionnels de Santé). The RPPS is the reference file of health professionals in the French health and social sector. It was developed by the state in collaboration with the recommendations of health professionals and the state health insurance. It is an exhaustive database that lists all identification data, diplomas, activity, and mode and structure of the practices of all health professionals. The four strata were defined according to the INSEE (French national institute for statistical and economic studies) rural/urban 2010 classification [34] and were: (1) rural zone with <2000 inhabitants; (2) urban zone 1 with 2000 to 9999 inhabitants; (3) urban zone 2 with 10,000 to 49,999 inhabitants; and (4) urban zone 3 with 50,000 or more inhabitants, which corresponded to the city of Rennes in Ille-Et-Vilaine. A sample of 425 GPs was selected with an allocation proportional to the number of 15–44 years old in Ille-et-Vilaine [35]- 27% in rural areas, 21% in urban zone 1, 13% in urban zone 2, and 39% in urban zone 3. Consequently, 115 GPs were drawn from the RPPS in the rural stratum, 90 in urban zone 1, 55 in urban zone 2, and 165 in urban zone 3. Three investigators contacted all the drawn GPs by telephone. For the second degree, study days were defined with the general practitioner so that an investigator can stay in the waiting room to include patients. The GP waiting rooms constitute an intermediate level between the GPs and patients/participants. Investigators stayed in waiting room on 15.5 Mondays, 8.5 Tuesdays, 1.5 Wednesdays, 8.0 Thursdays, 13.0 Fridays, and 0.5 Saturdays. The waiting rooms could be shared with other GPs in the practice; all eligible individuals entering the waiting room (patients and those accompanying them) were invited to participate. For the third degree, a sample of patients was drawn for each GP and each day. Within one day in a GP’s waiting room, we included two to 26 patients.

Finally, among the 115 GPs selected at random in the rural zone, 19 were not eligible because they were not or no longer GPs. Among the eligible GPs, 27 were contacted, of whom 14 participated. In their waiting rooms, 119 patients were eligible and 107 participated. In urban zone 1, among the 90 pre-selected GPs, 75 were eligible, 18 were contacted, and 11 participated. Among the 198 eligible patients, 162 participated. In urban zone 2, among the 55 pre-selected GPs, 20 were eligible, 11 were contacted, and five participated. Among the 61 eligible patients, 54 participated. In urban zone 3, among the 165 pre-selected GPs, 64 were eligible, 43 were contacted, and 20 participated. Among the 255 eligible patients, 231 participated. Then, the participation rate at the GP level was 50.5% (50/99) mostly due to their refusal (40/50). The participation rate at the patient level was 87.5%. Reasons for non-participation were no interest or no time (46.1%), the start of the medical consultation (16.7%), the language barrier (15.4%), a health reason (12.8%), and others reasons (9.0%).

### 2.2. Collected Data

While in the waiting room, patients completed a hand-delivered questionnaire to collect their socio-demographic characteristics and their awareness/perceptions, knowledge, and practices in terms of indoor home environment pollution. The average time required to complete the questionnaire was 15 min.

Awareness/perceptions were evaluated through four questions: (i) “Have you ever heard of indoor pollution?” (yes/no); (ii) “At what level do you think the chemicals in your indoor environment are a risk for your health?” for which response was given using analog visual scales from 0 to 10; (iii) “In your opinion, are the chemical substances present in the environment so ubiquitous that they cannot be avoided?” (completely agree, agree, mostly disagree, totally disagree), which was taken from the article of Barrett et al. [36]; and (iv) “Concerning pollution of the indoor environment, do you consider that the risks for French people in general are low, medium, high, or very high? This was taken from the French Barometer 2017 of the “Institut de Radioprotection et de Sûreté Nucléaire” (IRSN) on the perception of risks and security [17].

There is currently no validated questionnaire in the scientific literature to assess individual knowledge of environmental risk in general. It was therefore necessary to create one specifically for this study. A consensus method, the nominal group, was used [37,38]. Based on this method, a meeting was held with eight environmental health experts to determine the most relevant questions for assessing knowledge. These experts comprised a mixed group of diverse ages and experiences from a variety of professions (researchers in environmental health, general practitioner, midwife, therapeutic education nurse). Each expert came to the meeting with 10 questions they considered to be relevant to assess the population’s knowledge of environmental pollution and health. After an iterative round table, 67 questions were proposed. After the meeting, experts gave each of the 67 questions a note from 0 to 10 according to their relevance and proposed rewording if necessary. Finally, questions with median note ≥7.5 and with the highest median note in the case of a very close question within the same theme was selected. The final questionnaire consisted of 11 questions (one open-ended question was not taking into account due to missing data):-Who do you think is most vulnerable to pollution? and were asked to rank the following categories from most vulnerable (1) to least vulnerable (4): fetus, infant, adult, and elderly person.-For the following statements, respondents were asked to determine whether the proposed sentences were true, false, or if they did not know how to answer:
◦Smoking during pregnancy is not harmful to the health of the unborn child.◦If the person smoking in the home is not in the same room as the child, there is no second-hand smoke for the child.◦Diesel particulate matter increases the risk of cancer.◦Farmers, who are highly exposed to pesticides, are more likely to develop Parkinson’s disease.◦A natural product is harmless.◦Brittany is an area of high exposure to radon.◦Food containers may contain harmful substances.◦Drinking tap water is dangerous to your health.◦Is the air breathed in all dwellings polluted?◦Did you find it useful to reduce the use of cosmetics during pregnancy?

Finally, a score was calculated from these 11 questions, which allowed the assignment of a score between 0 and 11 to each respondent. A point was awarded for correct answers and 0 for incorrect answers or “I don’t know” or no-response.

The questions on practices were taken from environmental questionnaires already created for cohorts interested in environmental health such as PELAGIE or EDEN cohorts, but also from brochures proposed by perinatal network of Ille-et-Vilaine [24] or the guide on indoor air pollution of Santé Publique France (SPF) agency [20] as well as various scientific articles [9,31,36,39]. In total, 26 questions explored practices in the following areas: air renewal, combustion, hygiene and cosmetics, textiles, housekeeping, furniture and decoration, do-it-yourself, and construction (Document S1). 

### 2.3. Ethics

This study received a favorable opinion from the Ethics Committee of the Rennes University Hospital on 4 December 2018 (n° 18.93). 

An information sheet was given to each participant before completing the questionnaire, informing them of the subject of the study, its modalities, and the way the information collected was to be processed as well as its entirely confidential nature, in accordance with the law of 6 January 1978 on Information Technology and Freedom. The individuals participating in the study were also informed of the possibility of exercising their right of access to information concerning them for possible corrections or deletions of data by contacting the scientists in charge of the study.

### 2.4. Statistics

Weights were assigned to each participant to account for the sampling plan and thus the different probability of them being included. Weights were calculated within each stratum and associated: (i) a GP weight, which is the RPPS database size divided by the effective size of the sample of GPs, corrected for the proportion of non-eligibility and non-participation of GPs; (ii) a waiting-room weight, which accounted for the fact that waiting rooms could be shared with other GPs or not (a weight-sharing method was used at this step [40] ); and (iii) a participant weight, accounting for the participation rate. Thus, all the data (excepted effectives) presented account for the sampling design and weighting. Then, quantitative variables were described by the participant and the mean ± standard deviation, and the qualitative variables by the participant and percentage. To explain perceptions (qualitative variables) and practices (qualitative variables) according to socio-demographic characteristics and with each other, multiple simple logistic regressions were performed for binary variables and multinomial logistic regression for variables of more than two modalities. Mean comparison tests were performed using Student or ANOVA tests to describe knowledge according to socio-demographic characteristics, perceptions, and practices. All statistical tests had a significance threshold of 0.01. Statistical analyses were performed using SAS software, v.9.4 (SAS Institute, Cary, NC, USA).

## 3. Results

### 3.1. Description of the Population

The study participants were predominantly female and the average age was 33.4 years (±22.3 years). Almost half of the participants reported working as employees (249/554), 15.2% (92/554) were in management or higher intellectual occupations, and 13.7% (95/554) were unemployed. The remainder had an intermediate occupation (42/554) or were laborers (37/554), craftsman/craftsman/trader/business managers (31/554), or farmers (5/554). Most reported having children and 49 respondents were expecting a child when the questionnaire was completed. The average number of children per respondent was 1.4 (±4.0) (Table 1).

### 3.2. Awareness and Perceptions

#### 3.2.1. Level and Source of Information

In total, 396 people (71%) had already heard of pollution of the indoor environment. Everyone answered this question. Those who had heard of this subject were older, mostly women, and were not expecting a baby (Table 2).

Among the 396 respondents who had heard about indoor environmental pollution, 160 (41.9%) said they were poorly informed, 180 (43.2%) moderately informed, 50 (13.5%) well informed, and six (1.4%) very well informed. Ninety-seven (25.8%) considered their level of information to be insufficient.

Health professionals were cited in nine (2%) of 352 responses as the source of information. The doctor’s office was cited four times (0.8%). The most frequently cited sources of information by participants were television (90/352, 24.2% of responses), media without more precision (74/352, 20.1%), Internet (65/352, 19.8%), family and friends (27/352, 7.4%), radio (18/352, 5.4%), and professional environment (13/352, 2.5%). The following were also mentioned: associations, social networks, documentaries and reports, newspapers, various readings, press, magazines, studies, personal research in the context of housing (renting, buying, works), information, posters, brochures, awareness campaigns, general knowledge, mail, advertising, an expert, personal experience, and packaging of cleaning products.

To the question “in your opinion, how would you like this information to be passed on to you” with the suggestions “brochures, health professionals, media, others”, 167/396 respondents (39.0%) wanted to be informed by health professionals, 201/396 (49.7%) by media, and 225/396 (58.7%) by brochures.

#### 3.2.2. Perception of Individual and Collective Risk

The mean score on the analog visual scale to the question “at what level do you think the chemicals in your indoor environment are a risk for your health?” was 6.5 ± 7.9. Three of four respondents scored between 5 and 10. The data were missing for 17 respondents. Concerning the statement “In your opinion, are the chemical substances in the environment so ubiquitous that they cannot be avoided?”, 2.6% totally disagreed, 19.3% mostly disagreed, 51.9% agreed, and 26.2% completely agreed. The data were missing for eight respondents. Concerning the risks for French people in general, on a risk scale ranging from low, medium, high, to very high, 7.6% of respondents considered that pollution of the indoor environment represents a very high risk for French people, whereas 15.6% considered that airborne pollution represents a very high risk. These answers are illustrated in Figure 1. The data were missing for 8 and 9 respondents, respectively, for these two questions.

Perceptions according to socio-demographic characteristics are illustrated in Table 3. More women strongly agreed, versus totally or mostly disagreed, (OR 2.91 [1.36–6.22]) than men to the statement that chemical substances in the environment are so ubiquitous that they cannot be avoided. They were also more numerous in considering that pollution of the indoor environment constitutes a high or very high risk for French people versus low or medium risk than men, but this association was not significant. Having one or more children was associated with a higher frequency of considering that pollution of the indoor environment constitutes a high or very high risk for French people (OR 1.90 [1.06–3.41]).

### 3.3. Knowledge

The knowledge score was calculated for 554 participants and was 6.6 ± 6.6. Scores ranged from 0 to 11, which were the minimum and maximum scores possible. The median was 7 and half of the participants had a score between 5 and 8. Just over half (57.4%) ranked infants and fetuses as the most vulnerable to pollution relative to adults and the elderly. Active smoking during pregnancy was known to be dangerous by most respondents (89.4%) and the concept of passive smoking was also well known (80.7%). The vast majority (86.8%) also knew that food containers could contain toxic substances. Almost half (48.6%) thought that the air in all homes was polluted. Slightly more than half (66.1%) thought that it could be helpful to decrease the use of cosmetics during pregnancy and 52.0% disagreed that drinking tap water was bad for health. A total of 52.1% said “a natural product is harmless” was false. A large majority (73.9%) did not know whether Brittany was an area of high exposure to radon or not (Figure 2).

People living in couples had a significantly higher knowledge score (6.8 ± 6.7 for people living in couples versus 6.0 ± 6.1 for people who were not, *p* < 0.0001). A higher level of education was also associated with a higher knowledge score (7.7 ± 6.1 for Master’s level or more, 7.0 ± 5.7 for 2 or 3-year university level, 6.2 ± 5.9 for French secondary school diploma, and 5.6 ± 7.2 for people having less than a French secondary school diploma, *p* < 0.0001). There were no significant differences associated with age, sex, degree of urbanization of the residence, or having or expecting a child.

### 3.4. Practices

#### 3.4.1. Air Renewal

Forty percent (230/553) of the study population opened windows for more than 10 min more than four times a week in the winter (versus three times a week or less) versus 85.0% in the summer. Controlled mechanical ventilation (CMV) was the most frequently used mode of ventilation found in dwellings (80.2%) and the ventilation system was cleaned at least once a year by 53.1% of the respondents (519/554). The average indoor temperature was 19.9 °C ± 5.2, with 92.5% of respondents reporting heating between 18 and 22 °C (480/554).

None of the socio-demographic characteristics were associated with home aeration in the winter. Fewer participants with a Master’s level degree or more had their CMV systems cleaned at least once a year than those with less than a French secondary school diploma (OR 0.20 [0.10–0.39]). Older participants were less likely to heat their homes above 22 degrees (OR 0.92 [0.85–0.99]). No other association were observed (Appendix A).

#### 3.4.2. Combustion

##### Tobacco

Concerning tobacco, 86.2% of the population surveyed (462/552) refused to allow people to smoke inside their homes, but when they did, it was at least four times per week in almost half of the cases (47.6%).

Living in a couple (OR 0.26 [0.12–0.56]) or having one or more children (OR 0.25 [0.1–0.67]) were associated with a decreased risk of allowing people to smoke in their home.

##### Others Sources of Combustion

Almost two-thirds of respondents (327/541) had their home heating system checked by a professional at least once a year. Among those with a chimney or stove (189/554), 94.1% had it swept once a year. The vast majority of respondents cooked in a pan at least once a week: one to three times a week for 45.5%, and four or more times a week for 42.3%. However, only half used a hood (50.9%) or ventilated the room (53.5%) regularly (versus never/rarely or occasionally) while cooking. The use of scented candles was rare, with 6.4% of respondents reporting lighting them one to three times per week and only 2.1% four or more times per week. The trend was the same for incense, with only 4% of people burning incense one or more times per week (2.8% one to three times and 1.2% four or more times).

Living in a couple (OR 1.75 [1.05–2.94]) or having one or more children (OR 2.9 [1.65–5.09]) were also associated with an increased likelihood of checking their heating system at least once a year (Appendix A). There were no other associations with the socio-demographic characteristics of the participants.

#### 3.4.3. Hygiene and Cosmetics

Slightly more than a third of respondents (37.2%; 208/553) replied that they checked the composition of the hygiene and cosmetic products they used and only 9.2% (61/551) used an application to select them. On average, the number of cosmetic products (excluding soap, shampoo, and toothpaste) used daily per person was 5.0 ± 9.7. This ranged from a minimum of 0 products to a maximum of 14 products. The average was 5.8 ± 9.6 for women and 2.8 ± 5.3 for men.

People with a Master’s degree or higher were more likely to use an application to select their hygiene products than those with less than a French secondary school diploma (OR 3.71 [1.56–8.82]). Women were more likely than men to use five or more cosmetic products daily (OR 8.30 [3.75–18.34]). The other results were not significant (Appendix A). There were no other associations with the socio-demographic characteristics of the participants.

#### 3.4.4. Textiles

Few people declared cleaning their clothes regularly at a dry cleaner, as only 3.9% (26/552) of respondents answered positively to this question. More than half (59.8%, 327/551) of the respondents replied that they regularly (versus never/rarely or occasionally) washed newly-bought textiles before wearing them and if they did not do so, 30.3% aired them. Moreover, most respondents said they used an industrial detergent (59.4%, 339/550), 29.4% (148/550) bought a so-called “eco-label” detergent, and 7.3% (339/550) made their own detergent. Almost half used fabric softeners (47.4%), 53.0% stain removers, and 18.0% bleach to care for their laundry.

Living in a couple was associated with a higher frequency of washing textiles regularly before first use than people living alone (OR 1.66 [1.013–2.711]). The type of laundry detergent used correlated with education level: more people with a French secondary school diploma and people with a Master’s level or more used homemade or eco-labelled laundry than those with less than a French secondary school diploma (OR 3.360 [1.55–7.26] and 5.569 [1.47–21.12], respectively) (Appendix A). There were no other associations with the socio-demographic characteristics of the participants.

#### 3.4.5. Housekeeping

Less than a third of the respondents said they regularly (versus never/rarely or occasionally) looked at the precautions for use (28.8%; 171/544) and the pictograms (27.4%; 149/541) of the products used. A small number (7.2%, 42/548) of participants regularly mixed several cleaning products at the same time, 16% (92/548) occasionally, and 73.7% (395/548) never or rarely. Less than half of the respondents (40.3%) removed dust from their dwellings by damp cleaning. Two thirds (66.5%) of the population used bleach: 23.5% (126/547) once a week maximum, 6.4% more than once a week (43/547), 36.6% (206/547) once a month or less, 30.5% (156/547) never, and 3.1% (16/547) did not know how to answer. Just over one quarter of respondents (26.3%) reported using an air freshener in their bathroom several times a week. 

Being older was associated with a decreased risk of regularly mixing cleaning products (OR 0.94 [0.89–0.98]) (Appendix A). There was no association between the frequency of reading the precautions for use and socio-demographic characteristics.

#### 3.4.6. Furniture and Decoration

After purchasing new furniture, 61% of participants reported airing their room as often as usual, 25.9% more often than usual only if there was an odor, 10.9% more often than usual, even if there was no odor, and 2.3% less often. There were no associations with their socio-demographic characteristics (data not shown).

#### 3.4.7. Do-It-Yourself and Construction

People were asked what they would do if they had to repaint their child’s room. The vast majority of respondents said they would aerate the room more often than usual during the work (90.4%), 68.0% would aerate the entire dwelling, and 71.4% would air the room more often several weeks after the work was completed. Most of the respondents (77.0%) said that children would not be allowed to enter the room while the work was in progress and 64.6% said that pregnant women would not be allowed to enter the room. Slightly more than half (58.6%) of the respondents looked at the composition of the paints used, but only slightly more than a third looked at the pictograms (38.3%) or wore a mask (37.4%) during the renovation of the room.

Being older was associated with less aeration during or after work less verification of the product composition (OR 0.96 [0.93–0.997]), less prohibition of allowing pregnant women (OR 0.94 [0.90–0.97]) or children (OR 0.96 [0.92–0.99]) to enter the room while the work was in progress. Having one or more children was associated with less prohibition of allowing pregnant women to enter the room while the work was in progress (OR 0.4 [0.21–0.78]) (Appendix A).

### 3.5. Associations between Practices, Knowledge, and Awareness/Perceptions

#### 3.5.1. Level of Awareness/Perceptions According to Knowledge 

Respondents who had already heard of indoor environmental pollution had a higher knowledge score than those who said they had never heard of it (6.9 ± 6.4 versus 5.9 ± 6.5; *p* < 0.001). There was no association between the knowledge score and perceptions explored by the questions “At what level do you think the chemicals in your indoor environment are a risk for your health?” and “In your opinion, are the chemical substances in the environment so ubiquitous that they cannot be avoided?”. However, there was a correlation between the level of perceived risk of the participants of indoor environmental pollution and the knowledge score (6.0 ± 5.3, 6.4 ± 7.0, 6.9 ± 6.0, and 7.2 ± 7.0 respectively for low, medium, high, or very high perceived risk, *p* = 0.007).

#### 3.5.2. Practices according to the Level of Awareness/Perceptions

There was no association between practices concerning air renewal and combustion and the level of awareness/perceptions of the respondents (Appendix A).

Concerning hygiene and the use of cosmetic products, people who had already heard about indoor environmental pollution and those who perceived a health risk for the French of indoor pollution to be high or very high more frequently used an application to choose their hygiene/cosmetics products. No association was observed between these practices and the perceptions explored by the questions “At what level do you think the chemicals in your indoor environment are a risk for your health?” and “In your opinion, are the chemical substances in the environment so ubiquitous that they cannot be avoided?”. There were no associations between the four perception items and checking the composition of hygiene products or the number of cosmetics used daily (Table 4).

In terms of textiles and housekeeping, having already heard about indoor environmental pollution was associated with more frequent washing of textiles before first use, washing textiles with homemade or eco-label laundry, and reading precautions for the use of cleaning products. No other associations were observed (Table 5).

Finally, concerning do-it-yourself and construction practices, those who had already heard about indoor environmental pollution were more likely to aerate if buying new furniture, but less likely to prohibit access of children to a room in which work was ongoing. Participants who perceived the health risk of indoor pollution to be high or very high for the French were more likely to aerate when buying new furniture. There were no other associations observed (Table 6).

#### 3.5.3. Practices according to Knowledge

There were eight practices for which the knowledge score was significantly higher: heating the home to no more than 22 °C, checking the composition of hygiene products, using an application to choose hygiene products, using fewer cosmetic products, washing textiles before first use, using homemade or eco-labelled detergent, never or rarely mixing different cleaning products, and aerating the home more often after buying new furniture. In addition, people who cleaned their CMV system had a significantly lower knowledge score (Table 7).

## 4. Discussion

### 4.1. Perceptions

Our results showed that adults of childbearing age in Ille-et-Vilaine were poorly informed about indoor environmental pollution. Twenty-nine percent of the study participants had never heard of it, whereas 10% had never heard of it in the French 2007 health barometer among the 6007 French people questioned [41]. However, these two studies had several differences: the age of the population studied (18 to 75 years for health barometer versus 18 to 45 years for ours), study location (throughout France for health barometer versus one French department in ours), and modality of interview (by telephone for health barometer versus hand-delivered self-administered questionnaire in ours). This may have resulted in differences in selection (inclusion by telephone was confronted with many refusals to participate, potentially from people who were perhaps uninformed and refused to answer out of disinterest) and differences in classification due to differences in the study modalities and age of participants (older participants are more likely to have heard about indoor pollution) possibly explain this discrepancy between the two studies. Twenty-six percent of respondents “strongly agreed” with the statement “chemicals are so ubiquitous that they cannot be avoided”. These results are similar to those of Barett et al. [36], who found that 25% of a cohort of pregnant women (mean age 31.3 years) who volunteered to participate and were interviewed by self-questionnaire strongly agreed with this statement. Once again, the results are difficult to compare due to study differences. Forty-two percent of respondents considered the risk due to indoor environmental pollution to the French to be “high” or “very high” versus 48.2% in the 2007 health barometer survey [41]. The question was slightly different, as it concerned “Indoor air pollution in buildings: homes, schools, offices” and not just the home and only indoor air and not the entire indoor environment. The perceived health risk of the participants was higher for outdoor pollution than indoor environmental pollution. This was also observed in the barometer study, with 84.4% of respondents perceiving a high or very high risk due to outdoor air pollution for the French versus 64.1% in our study [41]. Indeed, the impact of outdoor pollution on health is known and is classified as carcinogenic by the IARC (International Agency for Research on Cancer) [42]. This result is therefore logical. 

As in other studies, higher age [28,36,41] and education level [29,36,41] were associated with being better informed and having a better perception of the risks concerning indoor environmental pollutants. 

### 4.2. Knowledge

The proportion of correct answers varied greatly depending on the question, showing there to be areas of environmental risk that are still largely unknown to the general public. More than 80% of respondents correctly answered three questions (smoking during pregnancy is dangerous, food containers may contain harmful substances, passive smoking if someone is smoking in another room) and less than 50% correctly answered three questions (the air breathed in all dwellings is polluted, higher risk of Parkinson in farmers, higher radon exposure in Brittany). As already observed for perceptions, the public underestimated indoor air pollution. Although Parkinson’s disease appeared to be known by the public, its recognition as an occupational disease following exposure to pesticides was largely unknown, no doubt because this change is recent and concerns few professionals (2012) [43]. The fact that radon is a localized natural hazard and that current legislation does not require measuring radon levels for each dwelling may explain the lack of knowledge of this type of exposure [44]. The mean knowledge score was not very high, as it was 6.6 out of 11. There have not been many studies on this subject. Rouillon et al. [27] also observed a low score in 2017. They evaluated knowledge about endocrine disruptors in a population of 300 pregnant women with a 100-item questionnaire in an interview. The average knowledge score was 42.9/100. The difference could be explained by the fact that our questionnaire approached the environment in a global manner with simple and general questions, in contrast to the study of Rouillon et al., which dealt in depth with endocrine disruptors. In the 2007 health barometer, 67.2% of the respondents had already heard of the nine topics (excluding radon) and felt well informed about 5/9 topics on average [41]. This difference could be explained by the fact that the population surveyed was older and the questions asked were simply whether they were an environmental issue.

The level of knowledge was higher for those living as a couple and with a higher educational level. Better knowledge for couples can be explained by the fact that the experience and knowledge of the two individuals is additive. Rouillon also observed that a high level of education was associated with better knowledge of endocrine disruptors [28]. In the 2007 health barometer [41], being 18 to 25 years old and having less than a French secondary school diploma were associated with an increased risk of being under-informed. 

Better knowledge was associated with higher recognition of environmental chemicals as being potentially dangerous. The study of Rouillon et al. [28] also observed that a higher average risk perception score was associated with a higher level of knowledge.

### 4.3. Practices

Our results show certain recommended practices concerning indoor environment pollution were well integrated (by more than 80% of respondents). They were, in decreasing order: annual chimney and stove sweeping, maintenance of the temperature in the dwelling between 18 and 22 degrees, aeration of the dwelling during renovation, banning smoking inside the home, no-use of incense, and daily ventilation of the dwelling for more than 10 min in summer. Other practices were moderately implemented (by 60% to 80% of participants) and were, in decreasing order: prohibiting children from entering the room being renovated, not mixing cleaning products, ventilation of the renovated room several weeks after the work is completed, the ventilation of the entire dwelling during renovation work, not using scented candles, keeping pregnant women out of the room during works, and checking of the heating system. Finally, certain recommended practices were not well followed (by less than 60% of participants). They were, in decreasing order: washing textiles before first use, examining the composition of work and decoration products, annual cleaning of CMV systems, not using air fresheners, removing dust with a damp cloth, daily ventilation of the dwelling for more than 10 min in winter, paying attention to the composition of do-it-yourself products (38.3%), putting on a mask during renovation work, paying attention to the composition of cosmetic products, using homemade or eco-labeled detergent, airing the textile before the first use if it is not washed, paying attention to the precautions for use and danger pictograms for cleaning products, ventilating more often after buying new furniture to limit the accumulation of volatile organic compounds that escape from it, and using an application to choose hygiene products.

To date, studies to compare our results are rare. Daily ventilation was carried out in winter by 84.1% of the people questioned in the 2007 health barometer (compared to 40.0% in our study) [41]. Furthermore, 64.0% of people had not had their ventilation system checked in the last 12 months (compared to 53.1% in our study) [41]. This difference may be explained by the different methodologies used (general population versus primary care population, interviewed by telephone versus self-questionnaire, 18–75 years old versus 18–45). It is also important to pay attention to the composition of cosmetic products to reduce exposure to chemical substances that can have an adverse effect on fertility and embryogenesis. However, in the study of Cecile and al. [31], even though 54.8% of women believed there to be a risk in using cosmetics during pregnancy, very few changed their habits, except for nail polish and nail polish remover. In addition, in the regional survey “Indoor environmental health, behaviors and risks of exposure to indoor pollutants” conducted in Aquitaine in 2018, only 27.0% of women interviewed after leaving the maternity ward considered the existence of an eco-label as the main purchasing criterion for cosmetics, ahead of the brand and price [45]. To end, the dangerousness of new furniture due to their composition (agglomerates, glues, treatments, plastics, …) was little known by the population of our study, but also by that of the regional study conducted by the ARS of Aquitaine [45] since 35% of women interviewed considered that the composition of furniture had no real or no consequences on health against 7% for the paint.

The associations observed differed depending on the level of integration of a practice. Well-integrated practices were preferentially associated with socio-demographic characteristics (age, being in a couple, and having children). Only one (temperature in dwelling) was associated with knowledge. The under-integrated practices fell into two groups. On one hand, ventilation in winter, annual cleaning of CMV systems, and examining do-it yourself product composition showed no association for the first, and unexpected ones for the other two: cleaning CMV systems was more frequent when the level of education and knowledge were lower, and examining do-it yourself product composition was more frequent for younger participants. On the other hand, paying attention to the composition of cosmetic products or the use of an application, choice of detergents, precautionary use of cleaning products, and ventilation after the purchase of new furniture were mostly associated with knowledge and perceptions and the unique associated socio-demographic factor was the level of education.

### 4.4. Period of Vulnerability: Pregnancy and Childhood

Expecting a child was associated with an increased risk of never having heard of indoor environmental pollution. This finding is surprising, as pregnancy is a period of high vulnerability when expectant parents should be informed about environmental risks to the development of their future child. This lack of awareness among pregnant women has already been highlighted in previous studies: it has been spontaneously expressed by pregnant women in qualitative studies [46] and found in quantitative studies. Teysseire et al. [30] showed that 82% of women considered that they were not sufficiently informed about environmental risks and only 45% of the subjects were informed about environmental risks by a physician. Furthermore, Chabert et al. [29] showed that among 390 French women hospitalized in post-natal units, only a small proportion were informed about reprotoxic agents and their potential exposure during pregnancy, ranging from 6.8% to 39.3%, depending on the reprotoxic agent.

Having one or more children was associated with a higher frequency of considering pollution of the indoor environment to constitute a high or very high risk for French people. It is possible that parents are more sensitive to environmental risks because of the consequences they can have on their children’s health and more broadly on their children’s future lives. The French barometer study also showed that parents were significantly more likely to be dissatisfied with environmental health information, especially those of young children (under four years of age) [41].

### 4.5. Indoor Environment and Sex

To date, most studies on this subject have focused on women. According to our findings, being a woman was associated with a higher likelihood of having already heard of indoor environmental pollution. More women also fully agreed with the ubiquity of chemicals in the environment than men. These findings are in accordance with those in the literature, which show that gender is a dominant factor; women tend to express higher levels of concern about the environment than men [47]. The French barometer study [41] interviewed men and noted that women were significantly more likely than men to perceive environmental health risks as “high” or “very high”.

We did not observe any significant difference in knowledge between the sexes. We also did not observe any significant gender difference in practices, except for the number of cosmetics applied (more for women than men). There are no studies on the level of knowledge or practices among men with which to compare our results.

It would be worthwhile to inform men in the same way as women during a consultation or as a joint information campaign for couples.

### 4.6. Source of Information

The media (TV, Internet or other techniques of distribution of information) was very widely cited by respondents as the source of received information, whereas health professionals were only very rarely mentioned. This was shown by the study of Chabert et al. [29], which showed that better informed women were more likely to obtain the information themselves (Internet, media) and that of Teysseire et al. [30], in which 82.6% of pregnant women had received information via the Internet and 57% via television, whereas health practitioners were less frequently mentioned. However, in the aforementioned study [30], approximately one third of the women cited a health professional as one of their main sources of information, whereas they were cited by only 3% of our respondents. This can be explained by the fact that pregnancy is a period with close medical and para-medical follow-up, whereas our study surveyed all adults of childbearing age, men and women. In addition, our question was open-ended, asking for sources to be cited, whereas in other studies, the various sources of information were suggested, which increases the frequency with which they are mentioned.

Although few of our respondents had been informed by a health professional and many had been informed through the Internet, television, or the media, the distribution of responses concerning the desired source of information was more moderate: the media were highly represented (49.7%) but so were health professionals (39.0%). The importance of the media in our society no longer needs to be demonstrated, but this suggests that health professionals also have a role to play in providing information about environmental risks. Marie et al. [31] showed that only a minority of health professionals asked women during pregnancy about their exposure to chemicals and advised them to reduce exposure. There is thus room for improvement, especially as more than 65.0% of the women interviewed by Marie et al. [31] wished to be better informed by health professionals about the risks of using cosmetics, whether outside or during pregnancy. On this point, Rouillon et al. [28] have suggested that healthcare providers counsel pregnant women on exposure to environmental chemicals, while being careful not to increase their anxiety by advising them and taking into account their knowledge, perceptions, and possibilities for action. 

Of note, brochures were the most represented (58.7%) source of information, but again, this question was closed, whereas that concerning the media and health professionals was open and they had to be specified.

### 4.7. Strengths and Limitations

Our study had several strengths. The size of our sample was large (554), allowing more precise measurements. The method of recruitment by stratified random draw limited selection bias and allowed inclusion from different urban zones. The presence of the interviewers in the waiting room allowed good individual participant rates and correct completion of the questionnaires despite its length, since only seven questionnaires were excluded due to missing data. The questionnaire included questions drawn from previous studies and pre-existing official health recommendations. In addition, the questions concerning knowledge were defined by an expert group using a validated methodology.

Our study also had several limitations. The RPPS database is not completely up to date, as the GPs selected could have retired, are replacement doctors, or have a practice other than a general practice in a private practice. In addition, the participation rate of eligible GPs was approximately 50%, which may have introduced a selection bias if the GP’s participation is note random. Inclusions in stratum 3 were quite low. However, weighting allowed rebalancing between strata. In addition, three different investigators carried out the collection and could have influenced the recruitment due to their different personalities. However, the arguments were harmonized beforehand to eliminate such bias. The knowledge part of the questionnaire, created for this study using a validated method, was not validated, which may induce a lack of precision in the case of poor reliability or measurement bias in the case of poor validity. Finally, not all sources of domestic environmental pollutants were investigated including pesticides (which were, however, investigated through the “Pesti-Home” study) and food and water, as these are other routes of exposure.

## 5. Conclusions

Our study, which included both men and women, showed that certain practices for a healthy indoor environment are well integrated and others are not. The factors influencing the implementation of such favorable practices appear to differ depending on the frequency of their integration. Well-integrated practices were not related to knowledge, level of education, or perceptions, but rather to the responsibility of having a child. The implementation of less well-followed practices would be improved by better knowledge/information and a change in perceptions, for which an effort should be made to influence the practices of people from all socio-economic backgrounds. In line with recommendations made in the FIGO (International Federation of Gynecology and Obstetrics) statement [48], to avoid further increasing social inequalities in health, improvement of the population’s environmental health knowledge could be achieved through primary care professionals, who are the main contact of individuals within the healthcare system and the most likely to encounter populations vulnerable to environmental risk. Of course, primary-care professionals still need to be trained, and further studies are needed to develop key questions to identify the people most at risk and adapt the advice to be given to them.

## Figures and Tables

**Figure 1 ijerph-17-07669-f001:**
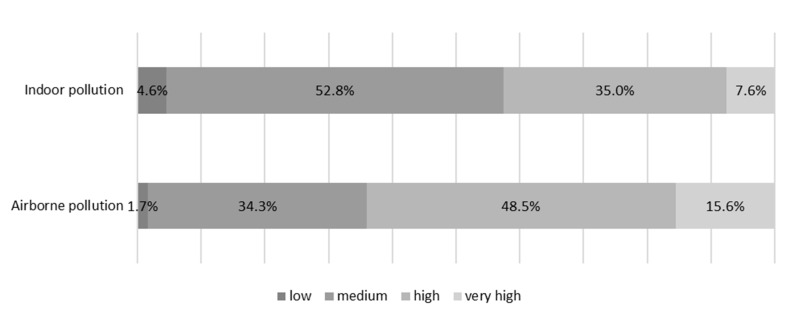
Perception of risks of the French on indoor pollution and airborne pollution.

**Figure 2 ijerph-17-07669-f002:**
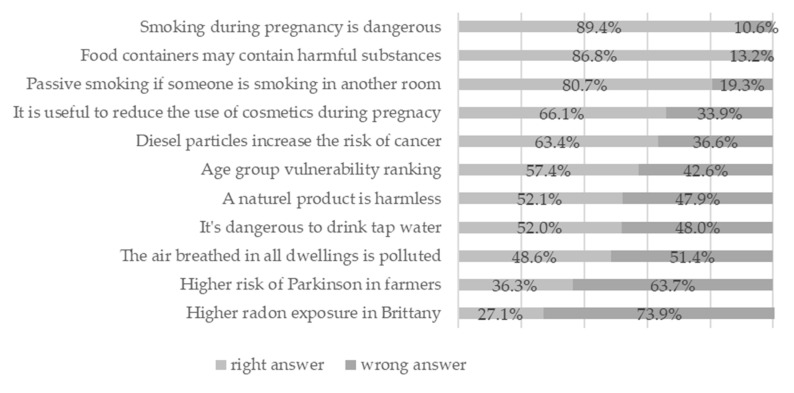
Details of answers to the knowledge questionnaire.

**Table 1 ijerph-17-07669-t001:** Population characteristics (*n* = 554).

Characteristics	Participants (%)
**Age (in years) ***	33.4 ± 22.3
**Sex**	
men	152 (28.4%)
women	402 (71.6)
**Degree of urbanization of their residence**	
Rural area <2000 people	107 (11.0)
2000 to 9999 people	162 (33.8)
10,000 to 49,999 people	54 (22.5)
≥50,000 people: Rennes	231 (32.7)
**Living in a couple ****	
no	188 (30.4)
yes	365 (69.6)
**Educational level ****	
Less than a French secondary school diploma	120 (23.7)
French secondary school diploma	149 (27.7)
Two- or three-year university level	170 (29.5)
Master’s level or more	110 (19.1)
**Having one or more children ****	
no	202 (33.3)
yes	351 (66.7)
**Expecting a child ****	
no	502 (89.7)
yes	49 (10.3)

* mean ± standard deviation, ** missing data: living in a couple and having one or more children *n* = 1, expecting a child *n* = 3, educational level *n* = 5.

**Table 2 ijerph-17-07669-t002:** Having already heard of indoor pollution according to socio-demographic characteristics (*n* = 554).

	Having Ever Heard of Indoor Pollution (yes) *	
Characteristics	OR [99% CI]	*p*-Value
**Age**	1.06 [1.01–1.11]	**0.001**
**Sex**		
men	1	**0.007**
women	1.92 [1.04–3.54]	
**Degree of urbanization of their residence**		
Rural area: <2000 people	1	0.917
2000 to 9999 people	1.07 [0.39–2.91]	
10,000 to 49,999 people	0.82 [0.16–4.19]	
≥50,000 people: Rennes	1.15 [0.47–2.81]	
**Living in a couple**		
no	1	0.567
yes	1.11 [0.67–1.84]	
**Education**		
Less than a French secondary school diploma	1	0.028
French secondary school diploma	1.38 [0.65–2.93]	
Two- or three-year university level	1.76 [0.68–4.56]	
Master’s level or more	3.56 [1.23–10.29]	
**Having one or more children**		
no	1	0.300
yes	1.3 [0.66–2.56]	
**Expecting a baby**		
no	1	**0.005**
yes	0.36 [0.14–0.91]	

* Reference: no.

**Table 3 ijerph-17-07669-t003:** Perceptions according to socio-demographic characteristics (*n* = 554).

	Level Perceived Risk of Indoor Chemicals *		Ubiquity of Chemicals **			High or Very High Perception of Risk of Indoor Pollution ***	
			Agree	Completely Agree			
Characteristics	Mean ± sd	*p*-Value	OR [99% CI]	OR [99% CI]	*p*-Value	OR [99% CI]	*p*-Value
**Age**			1.01 [0.96–1.07]	1.01 [0.94–1.07]	0.879	1.03 [1.00–1.06]	0.025
30 years old or less	6.3 ± 7.8	0.101					
More than 30 years old	6.6 ± 7.9						
**Sex**							
men	6.4 ± 8.3	0.673	1	1	**0.002**	1	0.062
women	6.5 ± 7.7		1.67 [0.95–2.92]	2.91 [1.36–6.22]		1.47 [0.86–2.53]	
**Degree of urbanization of their residence**							
Rural area: <2000 people	6.6 ± 5.8	0.933	1	1	**0.010**	1	**<0.001**
2000 to 9999 people	6.5 ± 8.4		0.57 [0.26–1.22]	0.98 [0.37–2.55]		1.09 [0.49–2.39]	
10,000 to 49,999 people	6.4 ± 11.2		1.17 [0.57–2.40]	2.28 [0.71–7.28]		0.57 [0.28–1.14]	
≥50,000 people: Rennes	6.6 ± 7.5		0.63 [0.33–1.19]	1.46 [0.69–3.08]		1.18 [0.57–2.47]	
**Living in a couple**							
no	6.3 ± 8.2	0.245	1	1	0.732	1	0.223
yes	6.6 ± 7.7		0.96 [0.36–2.60]	0.82 [0.33–2.00]		1.27 [0.75–2.17]	
**Education**							
Less than a French secondary school diploma	6.5 ± 8.7	0.190	1	1	0.138	1	0.088
French secondary school diploma	6.2 ± 8.4		0.96 [0.29–3.22]	0.91 [0.25–3.32]		1.38 [0.55–3.44]	
Two or three-year university level	6.6 ± 7.3		0.64 [0.23–1.74]	1.33 [0.36–4.95]		1.59 [0.73–3.47]	
Master’s level or more	6.7 ± 7.2		0.53 [0.16–1.71]	0.82 [0.24–2.82]		0.80 [0.31–2.09]	
**Having one or more children**							
no	6.3 ± 8.4	0.143	1	1	0.660	1	**0.005**
yes	6.6 ± 7.5		1.1 [0.46–2.66]	1.36 [0.55–3.41]		1.90 [1.06–3.41]	
**Expecting a baby**							
no	6.6 ± 7.8	0.282	1	1	0.711	1	0.019
yes	6.0 ± 8.9		0.91 [0.23–3.61]	0.59 [0.11–3.22]		0.39 [0.14–1.10]	

* Missing data: *n* = 17; ** Reference: totally disagree or mostly disagree. Missing data: *n* = 8; *** Reference: low or medium. Missing data: *n* = 8.

**Table 4 ijerph-17-07669-t004:** Practices concerning hygiene and cosmetics according to the level of awareness/perceptions.

	Check the Composition of Hygiene Products *		Use an Application to Choose Cosmetics Products *		Use Five or More Cosmetic Products Daily *	
	OR [99% CI]	*p*-Value	OR [99% CI]	*p*-Value	OR [99% CI]	*p*-Value
**Have ever heard of indoor pollution**						
No	1	0.015	1	**0.006**	1	0.183
Yes	2.61 [0.95–7.20]		3.46 [1.09–10.95]		0.75 [0.43–1.33]	
**Perceived level of health risk of indoor chemicals**						
≤4	1	0.474	1	0.681	1	0.572
≥5	1.27 [0.52–3.07]		1.16 [0.45–2.97]		0.81 [0.30–2.19]	
**Ubiquity of environmental chemicals**						
Disagree/mostly disagree	1	0.805	1	0.191	1	0.932
Agree	1.16 [0.57–2.37]		1.95 [0.57–6.66]		1.09 [0.54–2.19]	
Completely agree	1.02 [0.46–2.30]		2.54 [0.67–9.63]		1.11 [0.54–2.28]	
**Perceived health risk of indoor pollution by the French**						
Low or medium	1	0.134	1	**0.001**	1	0.450
High or very high	1.35 [0.80–2.28]		2.42 [1.23–4.76]		1.14 [0.72–1.79]	

* Reference: no.

**Table 5 ijerph-17-07669-t005:** Practices concerning textiles and housekeeping according to the level of awareness/perceptions.

	Wash Textiles Before First Use *		Wash Textiles with Homemade or Eco-Label Laundry *		Read Precautions for the Use of Cleaning Products **			Mixes Cleaning Products **		
					Occasionally	Regularly		Occasionally	Regularly	
	OR [99% CI]	*p*-Value	OR [99% CI]	*p*-Value	OR [99% CI]	OR [99% CI]	*p*-Value	OR [99% CI]	OR [99% CI]	*p*-Value
**Have ever heard of indoor pollution**										
No	1	**0.006**	1	**0.007**	1	1	**0.004**	1	1	0.114
Yes	1.70 [1.03–2.81]		2.31 [1.04–5.16]		1.86 [1.12–3.08]	2.23 [0.91–5.46]		1.10 [0.48–2.51]	0.53 [0.21–1.33]	
**Perceived level of health risk of indoor chemicals**										
≤4	1	0.057	1	0.029	1	1	0.023	1	1	0.358
≥5	1.73 [0.82–3.65]		2.45 [0.84–7.12]		2.54 [0.88–7.37]	1.82 [0.66–5.05]		1.13 [0.23–5.63]	0.57 [0.19–1.72]	
**Ubiquity of environmental chemicals**										
Disagree/mostly disagree	1	0.862	1	0.384	1	1	0.276	1	1	0.741
Agree	1.13 [0.62–2.04]		0.71 [0.36–1.39]		1.38 [0.76–2.49]	1.86 [0.75–4.63]		1.34 [0.54–3.31]	0.75 [0.13–4.31]	
Completely agree	1.17 [0.47–2.88]		0.78 [0.35–1.73]		1.31 [0.60–2.85]	1.81 [0.76–4.34]		1.23 [0.47–3.24]	1.05 [0.18–6.21]	
**Perceived health risk of indoor pollution by the French**										
Low or medium	1	0.343	1	0.503	1	1	0.257	1	1	0.160
High or very high	1.24 [0.68–2.24]		1.21 [0.57–2.53]		0.80 [0.43–1.47]	1.18 [0.58–2.39]		0.82 [0.47–1.43]	1.53 [0.57–4.13]	

* Reference: no; ** Reference: never/rarely.

**Table 6 ijerph-17-07669-t006:** Practices concerning furniture and works according to the level of awareness/perceptions.

	Aeration if Buying New Furniture *		Aeration More Often during/after Work *		Look at the Composition of the Products Used for Work *		Keep Pregnant Women out of the Room During Works *		Prohibit Access of Children to the Room During Works *	
	OR [99% CI]	*p*-Value	OR [99% CI]	*p*-Value	OR [99% CI]	*p*-Value	OR [99% CI]	*p*-Value	OR [99% CI]	*p*-Value
**Have ever heard of indoor pollution**										
No	1	**0.003**	1	0.116	1	0.192	1	0.020	1	**0.005**
Yes	1.88 [1.09–3.24]		0.17 [0.01–3.34]		1.38 [0.72–2.62]		0.58 [0.31–1.06]		0.44 [0.21–0.92]	
**Perceived level of health risk of indoor chemicals**										
≤4	1	0.110	1	0.391	1	0.408	1	0.426	1	0.351
≥5	1.60 [0.74–3.45]		0.44 [0.04–5.48]		1.30 [0.56–3.05]		1.21 [0.64–2.27]		0.73 [0.29–1.81]	
**Ubiquity of environmental chemicals**										
Disagree/mostly disagree	1	0.028	1	0.111	1	0.603	1	0.647	1	0.450
Agree	2.06 [0.72–5.89]		4.44 [0.40–48.80]		0.96 [0.56–1.63]		0.85 [0.35–2.04]		1.38 [0.69–2.77]	
Completely agree	2.74 [1.04–7.21]		0.69 [0.07–7.31]		0.80 [0.43–1.52]		0.67 [0.21–2.15]		1.14 [0.38–3.41]	
**Perceived health risk of indoor pollution by the French**										
Low or medium	1	**<0.001**	1	0.375	1	0.654	1	0.221	1	0.346
High or very high	1.87 [1.19–2.95]		0.50 [0.06–3.93]		0.90 [0.48–1.69]		1.31 [0.73–2.33]		1.21 [0.71–2.06]	

* Reference: no.

**Table 7 ijerph-17-07669-t007:** Practices according to knowledge score.

Practice	Mean ± sd *	Mean ± sd *	Mean ± sd *	*p*-Value **
**Home ventilation in winter per week**				
Response modality	<1 time	1 to 3 times	4 times or more	
Knowledge score	6.4 ± 6.6	6.8 ± 7.0	6.4 ± 6.1	0.050
**CMV cleaned at least once a year**				
Response modality	Yes	No		
Knowledge score	6.3 ± 6.8	6.8 ± 6.1		0.001
**Housing temperature**				
Response modality	<18°	18° ≤ T° ≥ 22°	>22°	
Knowledge score	7.7 ± 4.2	6.7 ± 6.6	5.7 ± 5.1	0.006
**Allowed indoor smoking**				
Response modality	Yes	No		
Knowledge score	6.4 ± 5.6	6.6 ± 6.8		0.531
**Checked heating systems at least once a year**				
Response modality	Yes	No		
Knowledge score	6.6 ± 7.1	6.6 ± 5.8		0.914
**Check the composition of hygiene products**				
Response modality	Yes	No		
Knowledge score	7.2 ± 6.5	6.2 ± 6.4		0.002
**Use an application to choose hygiene or cosmetic products**				
Response modality	Yes	No		
Knowledge score	7.4 ± 6.1	6.5 ± 6.6		0.002
**Number of cosmetics products used daily**				
Response modality	<5	≥5		
Knowledge score	6.9 ± 6.6	6.2 ± 6.4		<0.001
**Wash textiles before first use**				
Response modality	Never/occasionally	Regularly		
Knowledge score	6.3 ± 6.7	6.8 ± 6.5		0.002
**Laundry choice**				
Response modality	Eco-label/homemade	Industrial		
Knowledge score	7.4 ± 6.3	6.1 ± 6.3		<0.001
**Read precautions for use of cleaning products**				
Response modality	Never/rarely	Occasionally	Regularly	
Knowledge score	6.3 ± 6.4	6.9 ± 7.0	6.5 ± 6.2	0.091
**Mixed cleaning products**				
Response modality	Never/rarely	Occasionally	Regularly	
Knowledge score	6.8 ± 6.6	6.1 ± 6.3	5.6 ± 5.8	<0.001
**Aeration if buying new furniture**				
Response modality	Less than usual	As usual	More than usual	
Knowledge score	4.6 ± 5.6	6.5 ± 6.6	6.9 ± 6.4	0.003
**Aerate more often during/after work**				
Response modality	Yes	No		
Knowledge score	6.6 ± 6.6	6.5 ± 5.8		0.899
**Look at the composition of the products used for work**				
Response modality	Yes	No		
Knowledge score	6.8 ± 6.5	6.4 ± 6.7		0.033
**Keep pregnant women away from the work room**				
Response modality	Yes	No		
Knowledge score	6.6 ± 6.5	6.6 ± 6.9		0.959
**Prohibit access to the work room to children**				
Response modality	Yes	No		
Knowledge score	6.5 ± 6.6	7.0 ± 6.4		0.043

* sd: standard deviation. ** *p*-value of Student test for two modality variables. *p*-value of ANOVA for >2 modality variables.

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
