# Peer review of "Perceptions, Knowledge, and Practices Concerning Indoor Environmental Pollution of Parents or Future Parents"

_ijerph, 2020, doi:10.3390/ijerph17207669_

Round 1

Reviewer 1 Report

Thank you for the opportunity to review. This study examines perception, knowledge, and practices concerning indoor pollution among adults of childbearing age in Brittany, France in 2019.

Overall, the study is sound and thoroughly written. I only have a few suggestions to help improve the manuscript.

1) Although thorough, the paper is very long, and could benefit from shorteninng some parts, especially the results and Discussion.

2) On the other hand, it would be interesting to explore whether there are racial/ethnic disparities in perception, knowledge and practices. My hunch is some groups who may have different cultural or no access to information in their primary language. This may be contributing to their perception, knowledge and practices. The findings may be helpful to identify ways to raise awareness or provide education/training.

Author Response

Dear editor,

We have revised our manuscript in accordance with comments of the reviewers. Our answers to each comment are below the comment, in blue (following >>). Furthermore, we have made a supplemental file.

Sincerely,

Fabienne Pelé

Reviewer 1

Thank you for the opportunity to review. This study examines perception, knowledge, and practices concerning indoor pollution among adults of childbearing age in Brittany, France in 2019. Overall, the study is sound and thoroughly written. I only have a few suggestions to help improve the manuscript.

1 - Although thorough, the paper is very long, and could benefit from shorteninng some parts, especially the results and Discussion.

>> As suggested, we reduced some chapter taking into account remarks of others reviewers. In the results chapter we have delated data about housing characteristics (line 221-232). In the discussion chapter we have reduced the practices chapter (from line 498) and the strengths and limitations chapter (lines 618-631).

1 - On the other hand, it would be interesting to explore whether there are racial/ethnic disparities in perception, knowledge and practices. My hunch is some groups who may have different cultural or no access to information in their primary language. This may be contributing to their perception, knowledge and practices. The findings may be helpful to identify ways to raise awareness or provide education/training.

>> We thank the reviewer for this comment. Unfortunately, we do not ask for the ethnicity in the questionnaire and we excluded people who did not speak French. It is an interesting question which, I think may constitutes a subject in itself and probably requires specific methodology to properly questioned people whatever their culture or language.

Reviewer 2 Report

This article is well-written and provides some findings on the perception, knowledge, and practices concerning indoor environmental pollution.

Title: Is there any reason to emphasize quantitative study?

Introduction : It is generally known that the fertile period is 15 to 49 years old. Please provide the rationale for setting the study subject to be 18-45 years old.

Method: The significance of this study is to increase the perception, knowledge, and practice of indoor environmental pollution in adults at childbearing age. So, describe why the study was targeted to adults at childbearing age.

Method: In this study, tools to measure the concepts of perception, knowledge, and practice were all developed by the author. Please provide the reliability value of each tool, and if possible, add the contents to check the validity.

Author Response

Dear Reviewer,

We have revised our manuscript in accordance with your comments. Our answers to each comment are below the comment.

Sincerely,

Fabienne Pelé

This article is well-written and provides some findings on the perception, knowledge, and practices concerning indoor environmental pollution.

1 - Title: Is there any reason to emphasize quantitative study?

>> Indeed, there is no reasons to emphasize that this study is quantitative. We have modified the title (line 3).

2 - Introduction : It is generally known that the fertile period is 15 to 49 years old. Please provide the rationale for setting the study subject to be 18-45 years old.

>> This choice was based on ethical and methodological issues. Studies including minors are much more complicated to put in place in France due to ethics authorizations. This reason defined the lower age limitation. Concerning the higher limit age, it depend on the limits of the age classes used by the INSEE (French national institute for statistical and economic studies) in their description of the population structure which we used to determine the number of GPs to sample per strata. We agree with the reviewer about the fertile period, and we tried to get as close to it as possible. Regarding the distribution of first births according to the age of the mother in France in 2015 it is the case. Zero point height percent of births were to mothers under the age of 18 and none to mothers over the age of 45 (INSEE - Un premier enfant à 28,5 ans en 2015 : 4,5 ans plus tard qu’en 1974 – 2017 - https://www.insee.fr/fr/statistiques/2668280#tableau-figure2). We include INSEE reference in manuscript (lines 98, 99 and 103).

3 - Method: The significance of this study is to increase the perception, knowledge, and practice of indoor environmental pollution in adults at childbearing age. So, describe why the study was targeted to adults at childbearing age.

>> We targeted adults of childbearing age because the focus of this study is to increase the perception, knowledge, and practice of indoor environmental pollution in parents and future parents in order to improve environment for vulnerable individuals such as fetus and children. To be clearer we replaced “adults at childbearing age” by parents or future parents. We precise the title (line 4), some element in introduction (line 63-64) and in the objective (Lines 83-85) to be clearer.

4 - Method: In this study, tools to measure the concepts of perception, knowledge, and practice were all developed by the author. Please provide the reliability value of each tool, and if possible, add the contents to check the validity.

>> We do not developed all the tools. We developed only the questionnaire to evaluate knowledges. Unfortunatly, we do not evaluated neither the reliability nor the validity of this new questionnaire; this could be perform in another work. We add a sentence about that in the limits chapter (lines 643-645).

Reviewer 3 Report

The authors presented the results on perceptions, knowledge and practices regarding indoor environmental pollution using a stratified random sample of adults.

As a general comment, the discussion and conclusion are heavily focused on the situation in France. What would be the take-away for other countries? Are the results generalizable to other populations with different health systems and potentially different practices in the use of pollutants? I encourage the authors to add this information to the discussion and conclusion so that the results are if interest to more readers.

Additional Comments:

Abstract:

  • Line 26: do you mean “making sure” instead of “taking care”

Introduction:

  • Line 51: fix reference citation: [18-21]
  • Line 78: “making the waiting rooms of GPs accessible to all populate groups” is not clear. Do you mean to say that waiting rooms will therefore provide a representative sample ? Please revise
  • Line 81: “consulting in primary care”, replace with “consulting a primary care physician”

Materials ad methods:

  • Lines 94-95: add 3) and 4) to indicate the four strata
  • Line 104: do you mean “invited” instead of “proposed”
  • Line 105-106: can you elaborate on what you mean with “inclusions ranged from 2 to 26 over one day”. This last step remains unclear.
  • Line 128: “Which” change to This
  • Line 173- 174: add the 26 questions exploring the practices as a supplement file
  • Line 186: change “their” into “them” being included

Results:

  • Table 3: change “very agree” to strongly agree as is mentioned in the text
  • Table 3: What do the authors mean with “..of indoor pollution for French”, the title of the last column. Please clarify/ revise
  • Did the authors perform adjusted analysis for the perceptions reported in table 3? If not, what is the reason? This would be interesting to add.

Discussion:

  • Line 482- 504: This paragraph on the Practices repeats the results already mentioned in the Results, adding additional references. This is repetitive so please revise.
  • Line 599-612 explain that more respondents were female and had higher education levels and the potential reasons for this over-representation. If these patients are over-represented so not a representative sample of the population, it should be mentioned in the limitations section and not as a strength.

Author Response

Dear rewiever,

We have revised our manuscript in accordance with your comments. Our answers to each comment are below the comment following ">>". Furthermore, we have made a supplemental file. I will send it directly to editor as I can't do it here.

Sincerely,

Fabienne Pelé

The authors presented the results on perceptions, knowledge and practices regarding indoor environmental pollution using a stratified random sample of adults.

1 - As a general comment, the discussion and conclusion are heavily focused on the situation in France. What would be the take-away for other countries? Are the results generalizable to other populations with different health systems and potentially different practices in the use of pollutants? I encourage the authors to add this information to the discussion and conclusion so that the results are if interest to more readers.

>> We have modified the conclusion (lines 655-663)

2 - Abstract: Line 26: do you mean “making sure” instead of “taking care”

>> We have modified the manuscript (lines 26-27).

3 - Introduction:

Line 51: fix reference citation: [18-21]

>> We have modified the manuscript (lines 51).

Line 78: “making the waiting rooms of GPs accessible to all populate groups” is not clear. Do you mean to say that waiting rooms will therefore provide a representative sample ? Please revise

>> We agree with this remark, population present in GPs waiting room is not representative to the general population. This sentence is not clear. We have modified the entire paragraph according to the remark of another reviewer (lines 74-85).

Line 81: “consulting in primary care”, replace with “consulting a primary care physician”

>> We have modified the manuscript (line 84).

4 - Materials ad methods:

Lines 94-95: add 3) and 4) to indicate the four strata

>> We have modified the manuscript (lines 100 and 101).

Line 104: do you mean “invited” instead of “proposed”

>> We have modified the manuscript (line 110).

Line 105-106: can you elaborate on what you mean with “inclusions ranged from 2 to 26 over one day”. This last step remains unclear.

>> We have modified the manuscript to be clearer (lines 111 and 112).

Line 128: “Which” change to This

>> We have modified the manuscript (lines 134).

Line 173- 174: add the 26 questions exploring the practices as a supplement file

>> We have included those questions in supplemental file as requested.

Line 186: change “their” into “them” being included

>> We have modified the manuscript.

5 - Results:

Table 3: change “very agree” to strongly agree as is mentioned in the text

>> We have modified the Table 3 as requested, we replace very by completely as it is the word used in the text.

Table 3: What do the authors mean with “..of indoor pollution for French”, the title of the last column. Please clarify/ revise

>> We have deleted “for French” that is not necessary at this place and make confusion (Table 3 – last column).

Did the authors perform adjusted analysis for the perceptions reported in table 3? If not, what is the reason? This would be interesting to add.

>> We did not perform any adjusted analyses because our final objective is to use factorial and classification analysis to create profiles. These analyses are not yet performed and need another demand to my institution to be performed.

6 - Discussion:

Line 482- 504: This paragraph on the Practices repeats the results already mentioned in the Results, adding additional references. This is repetitive so please revise.

>> We have taken note of this remark. To make this chapter more readable we have removed references (which are note necessary we agree) and percentages. However, it seems to us that this chapter is necessary because it provides an overview of the practices carried out (Lines 498-519).

Line 599-612 explain that more respondents were female and had higher education levels and the potential reasons for this over-representation. If these patients are over-represented so not a representative sample of the population, it should be mentioned in the limitations section and not as a strength.

>> We thank the reviewer for this remark. Indeed, the comparison with the general population is not necessary since our population does not correspond to the general population and does not have to be representative of it. Our study population has to be representative of the population of people consulting a general practitioner. Which is the case. Indeed, It is well known that women consult more easily and more often general practitioner than men [1], particularly in this age group. Thus, they are consequently more represented than men in GPs waiting room. It is also well known that people from more precarious backgrounds consult GPs less often because access to care is more difficult for them [2]. Therefore, it is expected that the level of education in primary care is higher than in general population. We have delated this chapter (lines 618-631).

1 - Santé et recours aux soins des femmes et des hommes - Premiers résultats de l’enquête Handicap-Santé 2008 2008:8.

2 - Quels sont les éléments qui déterminent le recours aux soins. Centre d’observation de la société n.d. http://www.observationsociete.fr/sante/sante-sante/les-determinants-du-recours-aux-soins.html (accessed July 27, 2020).

Reviewer 4 Report

As general comment, it is an interesting work on a novel topic such as indoor pollution. Nevertheless, english should be improved. 

specific comments: 

Introduction

line 33: it seems that outdoor pollutants are different from heavy metals, OC etc. However, outdoor pollutants are also those listed. The sentence should be modified. 

line 43-44: the sentence "Environment is a mixture of physical, chemical, and biological pollutants that originate from outdoor air, building and decorative materials, combustion appliances, and human activity" should be modified. It should be the environmental pollution and not the environment which is a mixture of environmental pollutants. In addition, is it about indoor environment? or environment as a whole? It seems that it refers to indoor pollution, but it needs to be clarified. 

line 50: outdoor air pollution instead of air pollution.

line 62: start a new paragraph

line 73: I think teh paragraph is too long. I would just mention that since french population goes at least once a year to the GP and GP are a source of information about exposure to environmental pollutants, you selected thewaiting rooms of GP for recruitment. 

Methods:

2.1: I would change "Scheme and Study Population" for Recruitment Design and Study Population or something similar

line 88: start the paragraph with something similar to: the three-degree stratified random survey was designed as follows:

lines 115, 116 and 117: raison should be changed to reason

line 131: the explanation is too long, it could be shortened

Results:

Housing characteristics: if the housing characteristics are not used in any of the analyses performed in the study, it is not necessary to present them, it leads to confusion. 

line 235: define media, is it different from internet and television?

figure 1: improve the quality of the figure

line 301: I would separate tobacco smoking and the rest of the smoke sources. The risk perception of tobacco smoking is probably very different from the other smoke sources due to its great risk for health. 

Discussion:

line 508: the thrown question has not been answered

line 572: media, explain or specify

line 626: add water to possible routes. 

Author Response

Dear rewiever,

We have revised our manuscript in accordance with your comments. Our answers to each comment are below the comment following ">>".

Sincerely,

Fabienne Pelé

As general comment, it is an interesting work on a novel topic such as indoor pollution. Nevertheless, english should be improved. 

1- Introduction: 

line 33: it seems that outdoor pollutants are different from heavy metals, OC etc. However, outdoor pollutants are also those listed. The sentence should be modified. 

>> We have modified the sentence, we have forgotten a word “outdoor air pollutants” (line 33).

line 43-44: the sentence "Environment is a mixture of physical, chemical, and biological pollutants that originate from outdoor air, building and decorative materials, combustion appliances, and human activity" should be modified. It should be the environmental pollution and not the environment which is a mixture of environmental pollutants. In addition, is it about indoor environment? or environment as a whole? It seems that it refers to indoor pollution, but it needs to be clarified. 

>> We have precise the sentence. Indeed, some word were missing (lines 43 and 45).

line 50: outdoor air pollution instead of air pollution.

>> We have modified the manuscript (line 50).

line 62: start a new paragraph

>> We have modified the manuscript (line 63).

line 73: I think teh paragraph is too long. I would just mention that since french population goes at least once a year to the GP and GP are a source of information about exposure to environmental pollutants, you selected thewaiting rooms of GP for recruitment. 

>> We agree with this comment and have reduced this paragraph (lines 74-85).

2 - Methods:

I would change "Scheme and Study Population" for Recruitment Design and Study Population or something similar

>> We have modified the manuscript and noted “Study population” only (line 88). We think it is clearer.

line 88: start the paragraph with something similar to: the three-degree stratified random survey was designed as follows:

>> We have modified the manuscript (line 92).

lines 115, 116 and 117: raison should be changed to reason

>> We have modified the manuscript (line 121).

line 131: the explanation is too long, it could be shortened

>> We have reduced this chapter (lines 131-158).

- Results:

Housing characteristics: if the housing characteristics are not used in any of the analyses performed in the study, it is not necessary to present them, it leads to confusion. 

>> We agree and have removed this chapter (lines 221-232).

line 235: define media, is it different from internet and television?

The “source of information” question was open-ended with one or more answers. Each cited source was counted. For example, some participants noted media without any precision; others only TV or internet and others noted for example “TV, internet, media”, each of these sources/words were counted. We do not have more explanation of what participants want to say. For us media correspond to TV, written press and radio, internet. It is a technique for distribution of information. We have modified the manuscript to be clearer (lines 242-243).

figure 1: improve the quality of the figure

>> We have replaced the figure. We hope it will be better.

line 301: I would separate tobacco smoking and the rest of the smoke sources. The risk perception of tobacco smoking is probably very different from the other smoke sources due to its great risk for health. 

>> We have made two chapter as requested (lines 311-331).

Discussion:

line 508: the thrown question has not been answered

>> We have made correction to be clearer : “Daily ventilation, which is necessary to limit the concentration of pollutants inside the dwelling and to allow heating systems to work properly, was carried out in winter by 84.1% of the people questioned in the 2007 health barometer (compared to 40.0% in our study) [51]. This difference may be explained by the different methodologies used (general population versus primary care population, interviewed by telephone versus self-questionnaire, 18-75 years old versus 18-45). » (lines 520-534).

line 572: media, explain or specify

>> We have made precision in the manuscript (line 588).

line 626: add water to possible routes. 

>> We have modified the manuscript (line 647).
